# An Optical Measuring Transducer for a Micro-Opto-Electro-Mechanical Micro-g Accelerometer Based on the Optical Tunneling Effect

**DOI:** 10.3390/mi14040802

**Published:** 2023-03-31

**Authors:** Evgenii Barbin, Tamara Nesterenko, Aleksei Koleda, Evgeniy Shesterikov, Ivan Kulinich, Andrey Kokolov

**Affiliations:** 1Laboratory of Intelligent Computer Systems, Tomsk State University of Control Systems and Radioelectronics, 40, Lenin Ave., 634050 Tomsk, Russia; evgenii.s.barbin@tusur.ru (E.B.); ntg@tpu.ru (T.N.); kulinich@tusur.ru (I.K.); andrei.a.kokolov@tusur.ru (A.K.); 2Laboratory of Radiophotonics, Institute of Optics and Atmosphere, Siberian Branch of the Russian Academy of Sciences, 634055 Tomsk, Russia; 3Division for Electronic Engineering, National Research Tomsk Polytechnic University, 30, Lenin Ave., 634050 Tomsk, Russia

**Keywords:** MOEM-accelerometer, optical measuring transducer, inertial mass, movable and fixed waveguides, optical transmission coefficient, coupling length, threshold sensitivity, directional coupler, silicon-on-insulator (SOI)

## Abstract

Micro-opto-electro-mechanical (MOEM) accelerometers that can measure small accelerations are attracting growing attention thanks to their considerable advantages—such as high sensitivity and immunity to electromagnetic noise—over their rivals. In this treatise, we analyze 12 schemes of MOEM-accelerometers, which include a spring mass and a tunneling-effect-based optical sensing system containing an optical directional coupler consisting of a fixed and a movable waveguide separated by an air gap. The movable waveguide can perform linear and angular movement. In addition, the waveguides can lie in single or different planes. Under acceleration, the schemes feature the following changes to the optical system: gap, coupling length, overlapping area between the movable and fixed waveguides. The schemes with altering coupling lengths feature the lowest sensitivity, yet possess a virtually unlimited dynamic range, which makes them comparable to capacitive transducers. The sensitivity of the scheme depends on the coupling length and amounts to 11.25 × 10^3^ m^−1^ for a coupling length of 44 μm and 30 × 10^3^ m^−1^ for a coupling length of 15 μm. The schemes with changing overlapping areas possess moderate sensitivity (1.25 × 10^6^ m^−1^). The highest sensitivity (above 6.25 × 10^6^ m^−1^) belongs to the schemes with an altering gap between the waveguides.

## 1. Introduction

The accelerometer consists of a mechanical sensing element with an inertial mass and an optical electronic unit. The former converts acceleration into a displacement, while the latter senses the displacement. The accelerometers do not measure the acceleration signal directly, but by measuring the displacement of the inertial mass or the mechanical loads applied to the spring suspension system and induced by acceleration inertial forces. This can be achieved through various transformation methods, such as capacitive, piezoresistive, piezoelectric, thermal, optical, electromagnetic, tunneling effect, etc.

Micro-opto-electro-mechanical (MOEM) accelerometers are attracting growing attention due to the appreciable advantages of the sensors over their typical rivals. MOEM-accelerometers provide optical measurements via technologies used in microelectromechanical systems (MEMS) [1,2,3,4,5,6]. They are immune to electromagnetic noise, electrically insulated, corrosion–resistant and provide remote sensing and high sensitivity, which makes them preferable over rivals. These types of accelerometers cover many fields of application that state various requirements to their characteristics. For instance, in automotive electronics [7,8], the accelerometers are used to measure the accelerations in airbags, braking systems, electronic suspension and navigation systems. These applications do not require high accuracy of the accelerometers, while they state strict requirements of their size and cost. High bandwidth is critical for acoustic and vibrational measurements [9]. In an inertial navigation system, the accelerometer must generate low noise and possess fair stability at zero displacement [10]. For example, sensing acceleration in microgravity conditions requires extremely high acceleration sensitivity (less than µGal), long-term stability and uniform low-frequency characteristics [11]. The accelerometers play a crucial role in building monitoring systems. The measurement of building vibrations can detect defects and provide early warning. The accelerometers are used for monitoring seismic activity, drilling processes, high and low tides, volcanic events and other processes and operations [12,13,14,15,16]. Such accelerometers require superhigh sensitivity and a low-frequency response.

The principles of the optical measurement of displacement can be formed based on geometrical or wave optics [17]. The designs based on geometrical optics are simple and feature a high dynamic range, yet possess limited sensitivity due to their working principle [18,19,20,21].

In the wave-optics-based accelerometers, the acceleration alters the parameters of the light flux (phase, frequency, intensity, etc.). MOEM-accelerometers based on wave optics are tunnel, grating or interferometric resonators [22,23,24], Fabry-Perot resonators [25,26,27,28], photon crystals [29,30,31] and others. At present, MOEM-accelerometers based on the fiber Bragg grating (FBG) with direct integration into optical fiber are widely used [32,33,34,35].

Among the MOEM-accelerometer designs considered above, the highest variability, performance and applicability for measuring super low displacements are held by the systems based on the optical tunneling effect. The basic functional element with the highest practical potential in the optical transducer of a MOEM-accelerometer is the directional coupler. In the world literature, there are few studies describing the application of the optical tunneling effect for measuring acceleration. In particular, optical resonators are used to increase the sensitivity. For instance, Bhola [36] studies an accelerometer with a displacement sensitivity of 31 pm/g. Jian [37] presents an accelerometer with a *Q*–factor of 8.8 × 10^7^, sensitivity of 9 pm/g and measurement range of 130 g. F. Wan et al. constructed an accelerometer based on the Fano resonance in a ring resonator and interferometer, as per the Mach–Zehnder scheme, with a theoretical sensitivity of 111.75 mW/g [38]. G. H. Dushaq et al. measured the acceleration outside the sensor plane using a disk in a spring suspension mounted over a waveguide at a distance of 1 µm and achieved a sensitivity of 3 dB/g [39]. Table 1 summarizes the literature review. Evidently, the characteristics of the MOEM-accelerometers are given in different units, which is due to the choices of the respective authors.

Following Table 1, the optical measurement transducers cannot be adequately compared because the output values depend on the type of optical measuring transducer, the values of the inertial mass, transmission coefficients of the photodiodes and other optical and electrical components of the accelerometer. Therefore, they will be different for each device. This statement is particularly true for directional coupler-based schemes.

There are very few studies that compare various designs of accelerometers based on a directional coupler and the optical tunneling effect and analyze their strong and weak points and resulting characteristics. Hence, the analysis of the feasible designs of MOEM-accelerometers is a promising task.

The design of a MOEM-accelerometer starts with the choice of the functional scheme. Further, the scheme should be validated for its achievable characteristics and general feasibility. The experimental study of various schemes is extremely time- and cost-consuming due to the complexity of experimental specimen fabrication.

The aim of this study is to simulate and compare different functional schemes of MOEM-accelerometers featuring optical measuring transducers based on the directional coupler and the optical tunneling effect. This article presents possible designs of MOEM-accelerometers featuring different types of displacements of the waveguide with the inertial mass. We also present an analysis of the strong and weak points of the suggested schemes from the perspective of the complexity of the fabrication technology and anticipated sensitivity. This will provide minimal labor consumption for determining the concept of MOEM micro–g accelerometer fabrication using conventional technology.

## 2. Micromachined Sensing Chip

### 2.1. Sensor Fabrication

A MOEM-accelerometer includes three parts: mechanical, optical and electronic. To its simplest extent, an accelerometer can be represented as a “spring–mass–damping” scheme (Figure 1).

The accelerometer consists of the inertial mass *m* mounted in the housing on the spring suspension with stiffness *k*. The mass displaces relative to the fixed wafer under the acceleration *a*. The displacement *x* that is proportional to the acceleration is measured by an optical–electronic unit. The gap between the moving and fixed waveguides can be adjusted by the electrode structure that implements the electrostatic force *F_el_*.

The displacement of the inertial mass in relation to the fixed housing is provided by the force coupled with it, as per Newton’s second law:(1)$$F=m\cdot \left(a-\ddot{x}\right).$$

During displacement, the inertial mass is affected by the spring force of the suspension and the motion resistance force. Their sum equals *F* in Equation (1). As per Hooke’s law, the spring force is calculated as follows:$$F_{s}=-k\cdot x,$$
where *k* is the spring suspension stiffness.

The force of the inertial mass motion resistance in relation to the housing is:$$F_{r}=-\mu \cdot \dot{x},$$
where *μ* is the coefficient of the viscous damping forces.

The differential equation of the accelerometer’s inertial mass motion becomes the following:(2)$$m\cdot \ddot{x}+\mu \cdot \dot{x}+k\cdot x=m\cdot a+F_{el}.$$

Equation (2) can be transformed as follows:(3)$$\ddot{x}+\frac{\omega_{a}}{Q}\dot{x}+\omega_{a}^{2}x=a+\frac{1}{m}F_{el},$$
where *ω_a_* is the eigenfrequency of the accelerometer’s mechanical resonance, *Q* is the mechanical *Q*-factor of the accelerometer, $\omega_{a}^{2}x=\frac{k}{m}$ is the resonance frequency of the mechanical part of the accelerometer.

When the frequency of the measured acceleration is much less than the eigenfrequency *ω_a_* of the structure, the displacement *x* of the mass is proportional to the measured acceleration:(4)$$x=\frac{1}{\omega_{a}^{2}}a.$$

The optical subsystem includes a laser and an optical measurement transducer based on the optical tunneling effect and implemented as the directional coupler. The directional coupler is an optical “medium–air gap–medium” modulator that contains a fixed waveguide mounted on the housing and a movable waveguide mounted on the inertial mass. The waveguides are fabricated from silicon nitride [35,36,37] due to it possessing the lowest resulting losses compared to silicon photonics.

The design of a microaccelerometer is mainly determined by the technological process that is used for its fabrication. A MOEM-accelerometer can be made in two ways. The first method includes a one-step formation of interacting moving and fixed waveguides in a single plane (Figure 2). The base material is a silicon-on-insulator (SOI) wafer (Figure 2a). The waveguides are formed by depositing a silicon nitride layer on the device layer of the SOI wafer through a layer of dielectric material. The device layer of the SOI wafer is used to form the moving and fixed parts of the MEMS’s structure.

At the first stage, the mask for etching the handle layer of the SOI wafer is formed. Then, liquid etching of the silicon down to the SOI buried oxide (BOX) layer occurs (Figure 2b). Then, from the side of the SOI wafer device layer, silicon oxide (SiO_2_) and silicon nitride (Si_3_N_4_) are deposited to form the waveguides (Figure 2c). After that, the mask is formed and Si_3_N_4_ is etched down to the waveguide BOX layer (Figure 2d). At the next step, a SiO_2_ layer is deposited (Figure 2e) and the mask for SiO_2_ etching down to the device silicon layer is formed (Figure 2f). Then, the waveguide BOX layer is etched (Figure 2g) and deep etching of the SOI wafer device layer down to the SOI BOX layer is performed (Figure 2h). To release the moving mass, the passivation of the SOI wafer and liquid etching of the SOI BOX layer occur (Figure 2i). Figure 2j depicts a finished transducer after the passivating layer is removed. It is important to note that the working gap (less than 500 nm) formed between the waveguides, in this example of the technological process, exceeds the required value (by units of micrometers), and is then adjusted to the required value by the working gap adjustment system [39]. The fabrication of the transducer without the working gap adjustment system is possible, but comes with the complication of the technology. In this investigation, such a point is not critical and is not considered.

The second fabrication method includes the arrangement of the waveguides in different planes through layer-by-layer growing (Figure 3), with a SOI wafer as the base material (Figure 3a). In this case, the working gap between the waveguides is determined immediately by the thickness of the material layers.

First, the mask for etching the handle layer of the SOI wafer is formed. Then, liquid etching of the silicon down to the SOI BOX layer occurs (Figure 3b). After that, SiO_2_ and Si_3_N_4_ are deposited from the side of the SOI wafer device layer (Figure 3c), the mask is formed, and Si_3_N_4_ is etched to create the moving waveguides (Figure 3d). Next, the deposited SiO_2_ layer and moving waveguides are smoothed out, and the sacrificial layer of the photoresist is formed (Figure 3e). Subsequently, Si_3_N_4_ is deposited, and the fixed waveguide is formed through the photoresistive mask (Figure 3f). Then, the covering SiO_2_ layer is deposited and the photoresistive mask is used to open the windows to remove the photoresist sacrificial layer (Figure 3g). Figure 3h presents the transducer with a removed sacrificial layer of the photoresist and a column with the fixed waveguide. To release the moving part of the transducer, a mask on the other side of the SOI wafer is formed, and the consequent etching of the SOI BOX layer, device layer and SiO_2_ occurs (Figure 3i). Figure 3j depicts the finished transducer.

### 2.2. Functional Schemes of Accelerometer with Single-Plane Waveguides

Depending on the type of spring suspension, the moving waveguide, together with the inertial mass, can perform linear or angular movement along three axes. Thus, six types of functional schemes of MOEM-accelerometers can be built (Figure 4) with single-plane waveguides and different changing parameters of optical radiation. The first three schemes correspond to the linear displacement of the moving waveguide along axes X, Y and Z. In Schemes 4–6, the moving waveguide, together with the inertial mass, performs an angular movement along axes X, Y and Z. When measuring micro-g acceleration, the displacement of the waveguide with an inertial mass is measured in small values; hence, to high accuracy, we may assume that the angular movement of the waveguide is equivalent to its linear movement. Then, we may assume that in Schemes 1 and 4, the gap is changed; in Schemes 2 and 5, the coupling length is changed; in Schemes 3 and 6, the overlapping area between the moving and fixed waveguides is changed.

### 2.3. Functional Schemes of Accelerometer with Different-Plane Waveguides

The possible schemes of the MOEM-accelerometer with different-plane waveguides are presented in Figure 5. In Schemes 7–9, the moving waveguide, together with the inertial mass, performs a linear movement along axes X, Y and Z. In Schemes 10–12, the moving waveguide, together with the inertial mass, performs an angular movement along axes X, Y and Z.

When measuring the acceleration in Schemes 7 and 10, the overlapping gap between the moving and fixed waveguides changes; in Schemes 8 and 11, the coupling length between the waveguides changes; in Schemes 9 and 12, the gap between the waveguides changes.

## 3. Optical Measuring Transducer

### 3.1. Coupling Length

The optical measuring transducer (OMT) is part of the accelerometer, which is a MEMS structure with moving and fixed parts that have an air gap between them.

The threshold sensitivity of the accelerometer—the minimal signal that can be measured—can be quantitatively estimated as the noise–equivalent acceleration (NEA) in units of g/√Hz (*g* = 9.81 m/s^2^) [40,41].
(5)$$\mathrm{NEA}=\sqrt{a_{BR}^{2}+a_{L}^{2}+a_{PH}^{2}}.$$

The first term in the NEA expression is due to the thermal Brownian motion of the molecules in the accelerometer’s moving system. The rest of the terms in the NEA expression represent the laser (*a_L_*) and photodiode (*a_PH_*) noise.

The Brownian noise of the accelerometer is determined differently:

(a) for the accelerometer with linear displacement of the waveguide:(6)$$a_{BR}=\sqrt{\frac{4K_{b}T\omega_{a}}{mQ}}=\sqrt{\frac{4K_{b}T}{Q}\sqrt{\frac{k}{m^{3}}}},$$

(b) for the accelerometer with angular displacement of the waveguide:(7)$$a_{BR}=\sqrt{\frac{4K_{b}T\omega_{a}J_{y}}{mQ\cdot l}}.$$
where *K_b_* is the Boltzmann constant that equals 1.38 × 10^−23^ J/K; *T* is the absolute temperature, K; *J_y_*, and *l* are the moment of inertia and the displacement of the center of mass of the inertial mass in relation to the axis of its spring suspension.

According to Equations (6) and (7), the Brownian noise of the accelerometer can be reduced by increasing the mass of the moving system, reducing the resonance frequency and increasing the *Q*-factor [42]. A massive mechanical system is difficult to fabricate, while the reduction in the stiffness is less consuming.

At the same time, Equations (6) and (7) mean that the accelerometer with an angular movement of the waveguide presents √(m/l) times higher Brownian noise compared to the accelerometer with linear displacement at the same resonance frequency.

Considering the above, the schemes with angular movement have no advantages over linear schemes in terms of sensitivity. Therefore, let us determine the characteristics of the OMT only for the schemes with a linear displacement of the waveguides.

The choice of an optimal design of accelerometer requires studying the characteristics of the proposed schemes at various types of waveguide displacements and assessing the sensitivity (the slope of the optical transmission coefficient). The directional coupler—which is the basic element of the OMT—is characterized by the coupling length *L_cr_*, at which the optical power is completely transferred from one waveguide into another. The coupling length is calculated as per Equation (8) and corresponds to the minimal transmission coefficient at the output of the passthrough port [43].
(8)$$L_{cr}=\frac{\lambda }{2\Delta n_{eff}},$$
where *λ* is the wavelength (nm), ∆*n_eff_* is the difference between the effective refraction indices of even (*n_eff_*
_even_) and odd (*n_eff_*
_odd_) harmonics of the carrier mode in the waveguides. The dependence of the coupling length on the geometric dimensions of the single-plane waveguides are presented in Figure 6.

Following the figure, increasing the height and width of the waveguides will increase the coupling length. As the OMT is a MEMS structure with small gaps between the moving and fixed parts, the choice of the waveguide dimensions with lower coupling lengths is reasonable to reduce the probability of waveguides touching each other due to the skewing of the spring suspension caused both by the loads and fabrication process.

### 3.2. Characteristics of OMT

The external view of the OMT under investigation with different alignments of the waveguides is presented in Figure 7.

The optical transmission coefficient (*T_drop_*) is the relation of the transmitted optical power (*P_drop_*) at the output of the drop port to the input power (*P_input_*), and is determined as follows [44]:(9)$$T_{drop}=\frac{P_{drop}}{P_{input}}=\sin^{2}\left(\frac{\pi \Delta n_{eff}}{\lambda }L_{co}\right).$$

The dependencies of the optical transmission coefficient *T_drop_* for the drop port on the air gap *G* and coupling length *L_co_* for different fabrication technologies are presented in Figure 8. The calculations were made in the COMSOL Optics software package using the Finite-Difference Eigenmode (FDE) solver. The coupling length under study was limited to 100 µm due to the fabrication technology and to reduce the risk of moving and fixed parts of the MEMS structure touching each other; the air gaps under investigation were limited to 600 nm to reduce the crystal area and coupling length of the photonic integrated circuit (PIC).

According to the analysis, the technology of the different-plane waveguides allows the waveguide gap to be increased considerably while preserving the same coupling length. For instance, the required air gap for the OMT with a single-plane waveguide is 100 nm at a coupling length of 50 µm, while for different-plane waveguides, the gap amounts to 520 nm. For a coupling length of 50 µm and single-plane waveguides, the gap is 350 nm. The reduction in the waveguide width from 1000 to 750 nm—while preserving the same height—allows a reduction in the coupling length of the directional coupler or a further increase in the initial gap of the OMT.

#### 3.2.1. Changing the Coupling Length

In Functional Scheme 2 (Figure 4b) and Scheme 8 (Figure 5b), the inertial mass moves along axis X, and the coupling length of the directional coupler changes. Figure 9 presents the dependencies of the optical transmission coefficient on the coupling length. Evidently, the decreased gap reduces the period of the optical transmission coefficient sine. Hence, the sensitivity increases. From this point onward, to assess the sensitivity and the dynamic range of the obtained characteristics, let us choose the point that corresponds to a transmission coefficient of 0.5 and set it as the reference point for a corresponding axis. Figure 10 presents the dependencies of the transmission coefficient on the inertial mass movement along axis *X* at fixed gaps, where *L_co_* is the initial coupling length. For the waveguides with dimensions of 300 × 1000 nm, the sensitivity can vary between 12.5 × 10^3^ and 33 × 10^3^ m^−1^, while for the waveguide with dimensions of 300 × 750 nm, it can vary between 10 × 10^3^ and 83 × 10^3^ m^−1^ (Figure 10).

The same analysis was performed for Functional Scheme 8 (Figure 5b) with different-plane waveguides (Figure 11 and Figure 12).

According to Figure 11 and Figure 12, for the waveguide with dimensions of 300 × 1000 nm, the sensitivity may vary between 12.5 × 10^3^ and 250 × 10^3^ m^−1^, while for the waveguide with dimensions of 300 × 750 nm, it varies between 12.5 × 10^3^ and 500 × 10^3^ m^−1^.

According to the analysis of the various schemes of the accelerometer’s OMT, implying the movement of the waveguides along axis *X* and taking into account their fabrication technologies, the minimal sensitivity for each OMT type under study has identical slopes. An increased gap increases the sensitivity. An increased coupling length allows an increase in both the OMT’s dynamic range and the initial gap, while decreasing the sensitivity. In the case of OMTs with different-plane waveguides, the sensitivity can be adjusted in more ways as the optical coupling follows not the waveguide’s height, but its width. In addition, the different-plane waveguides are more flexible in the case of gaps exceeding 200 nm.

#### 3.2.2. Changing the Gap

For Functional Scheme 1 (Figure 4a) with single-plane waveguides, the inertial mass moves along axis Y. For Functional Scheme 9 (Figure 5c) with different-plane waveguides, the inertial mass moves along axis Z. The characteristics of the OMT with altering gaps are presented in Figure 13, Figure 14, Figure 15 and Figure 16. The coupling length *L_co_* remains the same.

Evidently, the movement of the waveguide with the inertial mass along axis *Y* (axis *Z* for different-plane waveguides) at a fixed coupling length changes the optical transmission coefficient according to the sine law. The sinusoid period decreases with the decreasing gap. An increased coupling length increases the number of sinusoid periods. A decreased gap increases the slope (Figure 14 and Figure 16).

For the waveguides with dimensions of 300 × 1000 nm, the sensitivity can vary between 6.25 × 10^6^ and 25 × 10^6^ m^−1^, while for the waveguide with dimensions of 300 × 750 nm, it can vary between 6.25 × 10^6^ and 50 × 10^6^ m^−1^. For the different-plane waveguides with dimensions of 300 × 1000 nm, the sensitivity can vary between 6.25 × 10^6^ and 250 × 10^6^ m^−1^, while for the waveguide with dimensions of 300 × 750 nm, it can vary between 6.25 × 10^6^ and 160 × 10^6^ m^−1^.

According to the analysis of the two types of waveguides and two different fabrication technologies, the minimal sensitivity for each of the OMTs under study has identical slopes. The increased coupling length and preserved sensitivity may provide an increased initial gap. In the case of OMTs with different-plane waveguides, the sensitivity can be adjusted in more ways as the optical coupling follows not the waveguide’s height, but its width.

#### 3.2.3. Changing the Overlapping Area

For Functional Scheme 3 (Figure 4c) with single-plane waveguides, the inertial mass moves along axis *Z*. For Functional Scheme 7 (Figure 5a) with different-plane waveguides, the inertial mass moves along axis *Y*. The characteristics of the OMT with altering overlapping areas are presented in Figure 17, Figure 18, Figure 19 and Figure 20. The coupling length *L_co_* and gaps remain the same. Evidently, the decreased gap increases the number of sinusoid waves. An increased coupling length allows an increase in the initial gap, while preserving the sensitivity. Increased sensitivity is possible at a considerably decreased initial gap.

For the waveguides with dimensions of 300 × 1000 nm, the sensitivity can be 1.25 × 10^6^ or 5 × 10^6^ m^−1^, while for the waveguide with dimensions of 300 × 750 nm, it can be 1.25 × 10^6^, 5 × 10^6^ or 10 × 10^6^ m^−1^.

For the different-plane waveguides (Figure 19 and Figure 20), the decreased gap increases the number of sinusoid periods, while the sensitivity weakly depends on the coupling length. For the different-plane waveguides with dimensions of 300 × 1000 nm, the sensitivity can vary between 1.25 × 10^6^ and 10 × 10^6^ m^−1^, while for the waveguide with dimensions of 300 × 750 nm, it can vary between 1.25 × 10^6^ and 12.5 × 10^6^ m^−1^. Table 2 summarizes the data for all of the functional schemes.

## 4. Discussion

From the perspective of fabrication, the MOEMS accelerometers with different-plane waveguides are more complex because they require more technological operations, including a complicated lift-off process. This reduces the percentage of usable samples and increases the final cost of the device. Moreover, the technology states high requirements for the flatness of the moving and fixed parts of the OMT.

The technology further complicates the case of feedback systems implemented as capacitive flat electrodes. In the case of single-plane waveguides, the feedback attenuators are much easier to implement as comb electrodes in the SOI wafer device layer.

The accelerometer pendulum in Schemes 9–12, with angular movement, can be fabricated by the same technology as the schemes with linear movement. They have no advantages over those with linear movement in terms of the sensitivity and dynamic range, yet they generate more Brownian noise, which raises the sensitivity threshold of the accelerometer.

The highest sensitivity (6.25 × 10^6^ m^−1^) belongs to the schemes with an altering gap between the waveguides. Functional Schemes 9 and 12, with different-plane waveguides in the case of the gaps, and an identical coupling length to those in Schemes 1 and 4, may feature higher sensitivity at gaps less than 100 nm (up to 500 × 10^6^ m^−1^). The dynamic range of all such schemes is limited to 80 nm, which appreciably hinders the practical application of the accelerometers that lack the feedback system for maintaining the inertial mass in place.

The schemes with changing overlapping areas (Schemes 3, 6, 7 and 10) possess moderate sensitivity (1.25 × 10^6^ m^−1^). The dynamic displacement range may reach ±400 nm. The technological complications, in this case, include the fabrication of different-plane waveguides. The interaction of the optical fields of the OMT waveguides also includes displacement, which may introduce additional modal distortion and losses.

The schemes with altering coupling lengths (Schemes 2, 5, 8 and 11) feature the lowest sensitivity, yet possess a virtually unlimited dynamic range, which makes them comparable to capacitive transducers. The sensitivity of the design depends on the coupling length and amounts to 11.25 × 10^3^ m^−1^ for a coupling length of 44 μm, and to 30 × 10^3^ m^−1^ for a coupling length of 15 μm. The sensitivity of these schemes can be increased by implementing the feedback systems.

The analysis presented in this article allows the choice of the accelerometer scheme that will implement the required characteristics. It should be noted that the final accelerometer’s sensitivity in units of pm/g, A/g or V/g will depend on the inertial mass, spring suspension stiffness, photodetector sensitivity and other parameters.

## 5. Conclusions

We have developed functional schemes of accelerometer OMTs based on the optical tunneling effect and have calculated their characteristics. Twelve OMT schemes with different positionings and displacements of the moving waveguide were considered. The schemes were compared in terms of their characteristics and fabrication technologies.

The results allow the choice of the accelerometer scheme with the required characteristics. Further studies will be aimed at the fabrication of the chosen accelerometer scheme and its experimental investigation.

## Figures and Tables

**Figure 1 micromachines-14-00802-f001:**
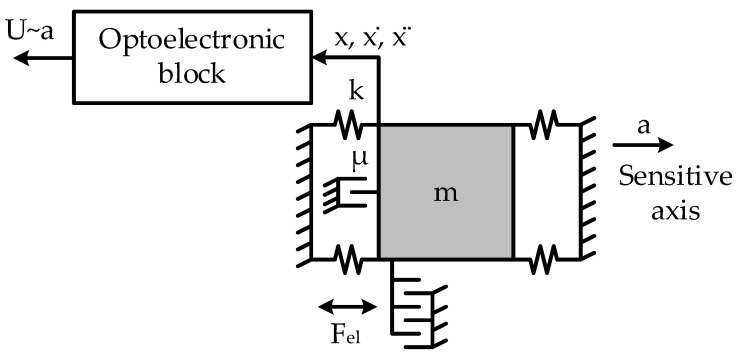
Functional scheme of an accelerometer.

**Figure 2 micromachines-14-00802-f002:**
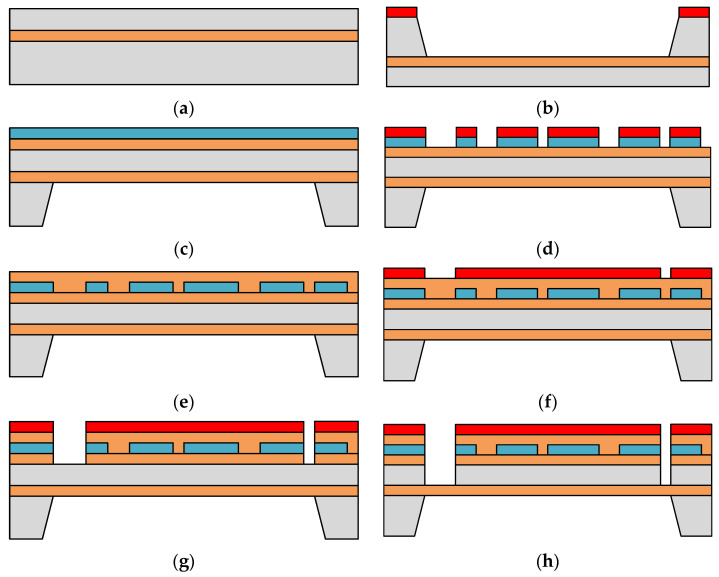
Technological process of a MOEM-accelerometer fabrication with single-plane waveguides. (**a**) SOI wafer; (**b**) Etching of SOI wafer handle layer; (**c**) Deposition of Si_3_N_4_ and SiO_2_ layers; (**d**) Etching of Si_3_N_4_; (**e**) Deposition of SiO_2_ covering layer; (**f**) Photoresist mask formation; (**g**) Etching of SiO_2_; (**h**) Deep etching of SOI wafer device layer; (**i**) Passivation of SOI wafer and SOI BOX layer removal; (**j**) Passivation layer removal.

**Figure 3 micromachines-14-00802-f003:**
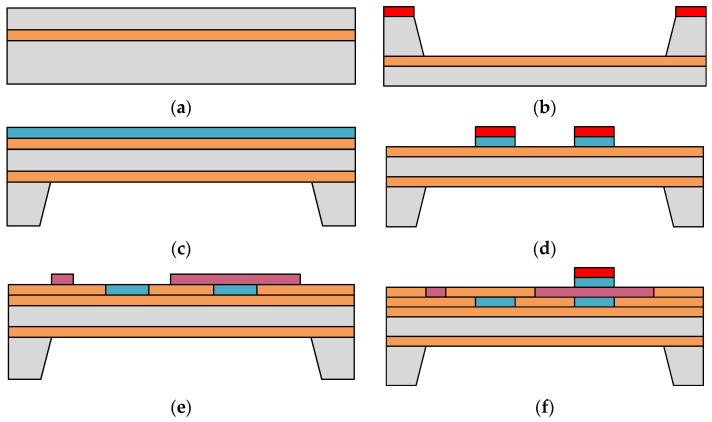
Technological scheme of MOEM-accelerometer fabrication with different–plane waveguides. (**a**) SOI wafer; (**b**) Etching of SOI wafer handle layer; (**c**) Deposition of Si_3_N_4_ and SiO_2_ layers; (**d**) Formation of moving Si_3_N_4_ waveguides; (**e**) Deposition and smoothing of SiO_2_, formation of photoresist sacrificial layer; (**f**) Formation of Si_3_N_4_ fixed waveguides; (**g**) Deposition of SiO_2_ covering layer; (**h**) Photoresist sacrificial layer removal; (**i**) Etching of SOI BOX layer, device layer and SiO_2_; (**j**) Photoresist removal.

**Figure 4 micromachines-14-00802-f004:**
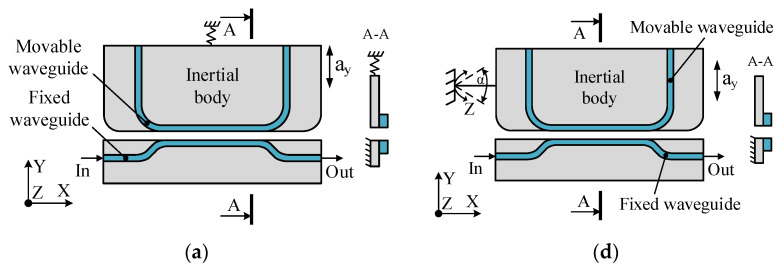
Functional schemes of the MOEM-accelerometer with single-plane waveguides. (**a**) Scheme 1, displacement along Y axis (gap change); (**b**) Scheme 2, displacement along *X* axis (length change); (**c**) Scheme 3, displacement along Z axis (overlap change); (**d**) Scheme 4, displacement around *Z* axis (gap change); (**e**) Scheme 5, displacement around *Z* axis (length change); (**f**) Scheme 6, displacement around X or Y axis (overlap change).

**Figure 5 micromachines-14-00802-f005:**
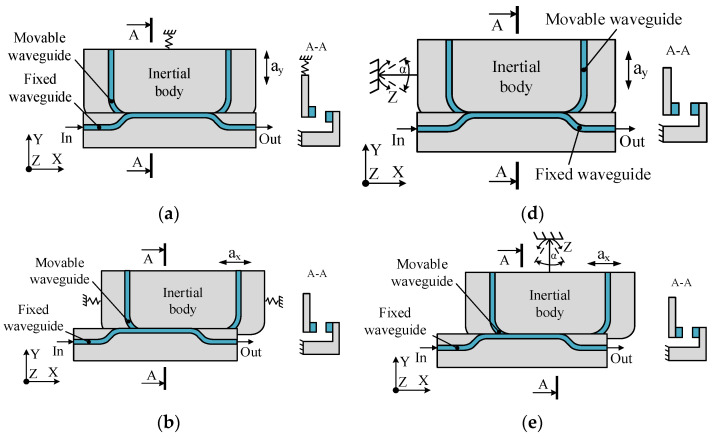
Functional schemes of the MOEM-accelerometer with different-plane waveguides. (**a**) Scheme 7, displacement along Y axis (overlap change); (**b**) Scheme 8, displacement along X axis (length change); (**c**) Scheme 9, displacement along Z axis (gap change); (**d**) Scheme 10, displacement around Z axis (overlap change); (**e**) Scheme 11, displacement around Z axis (length change); (**f**) Scheme 12, displacement around X or Y axis (gap change).

**Figure 6 micromachines-14-00802-f006:**
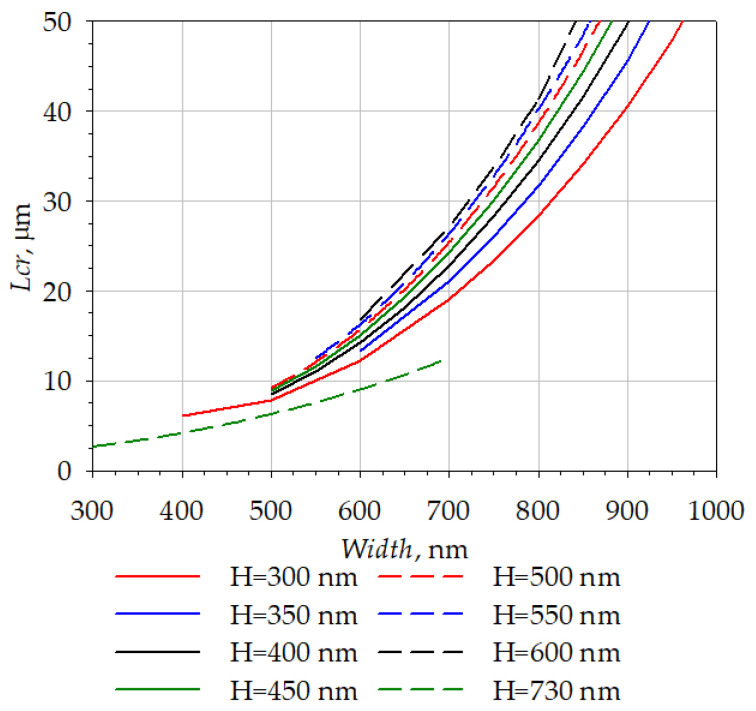
Dependence of the coupling length on the waveguide parameters.

**Figure 7 micromachines-14-00802-f007:**
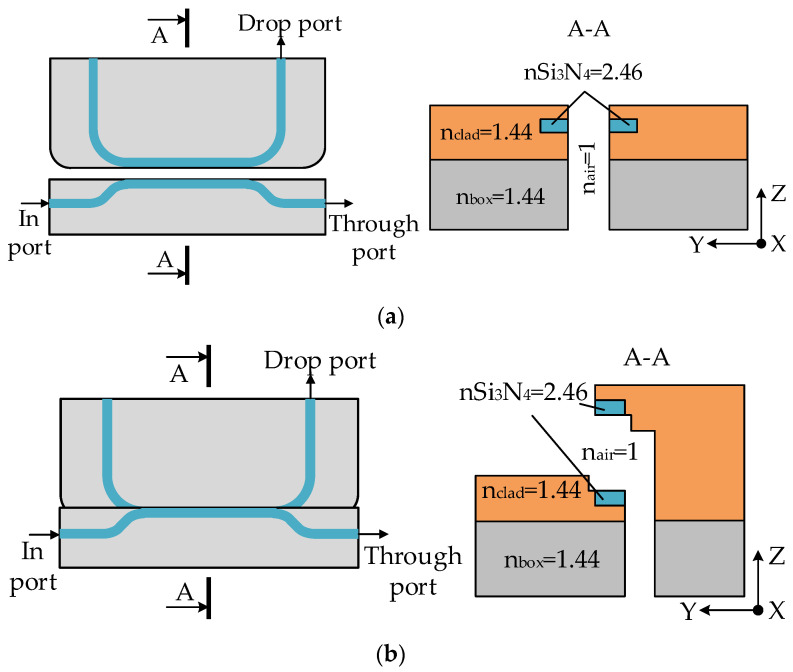
External view of the OMT. (**a**) Same-plane waveguides; (**b**) Different-plane waveguides.

**Figure 8 micromachines-14-00802-f008:**
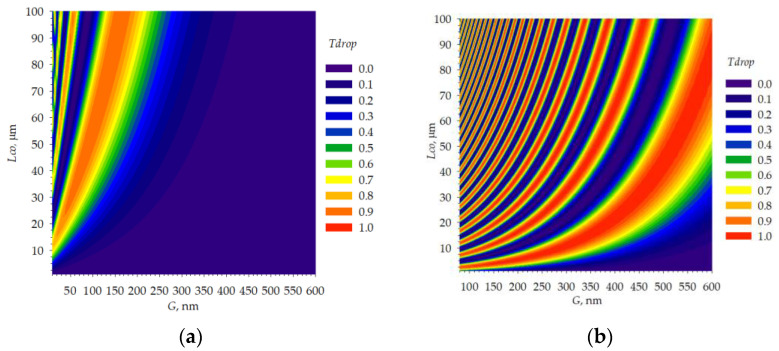
Dependence of the optical transmission coefficient of the directional coupler on the coupling length and gap. (**a**) Single-plane waveguides with dimensions of 300 × 1000 nm; (**b**) Different-plane waveguides with dimensions of 300 × 1000 nm; (**c**) Single-plane waveguides with dimensions of 300 × 750 nm; (**d**) Different-plane waveguides with dimensions of 300 × 750 nm.

**Figure 9 micromachines-14-00802-f009:**
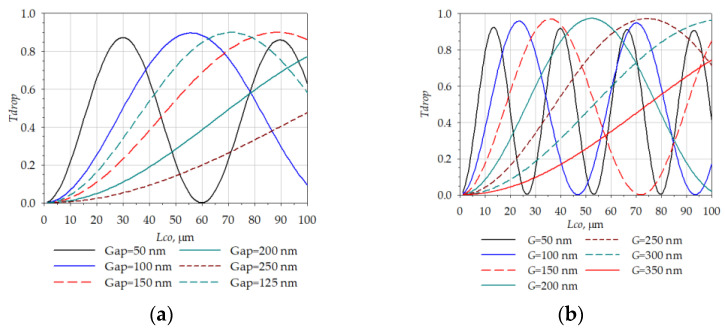
Dependence of the optical transmission coefficient on the coupling length for the OMT with single-plane waveguides (Scheme 2). (**a**) Waveguides with dimensions of 300 × 1000 nm; (**b**) Waveguides with dimensions of 300 × 750 nm.

**Figure 10 micromachines-14-00802-f010:**
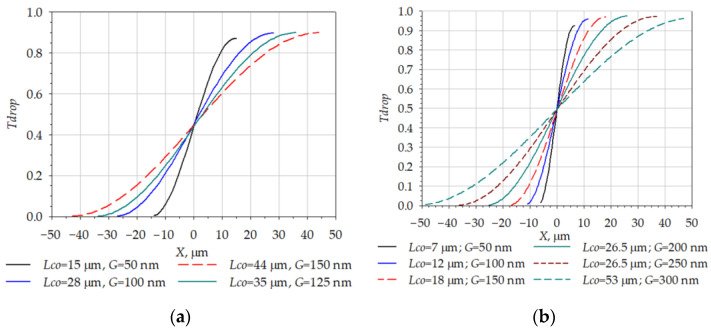
Dependence of the optical transmission coefficient on the inertial mass displacement along axis *X* for the OMT with single-plane waveguides (Scheme 2). (**a**) Waveguides with dimensions of 300 × 1000 nm; (**b**) Waveguides with dimensions of 300 × 750 nm.

**Figure 11 micromachines-14-00802-f011:**
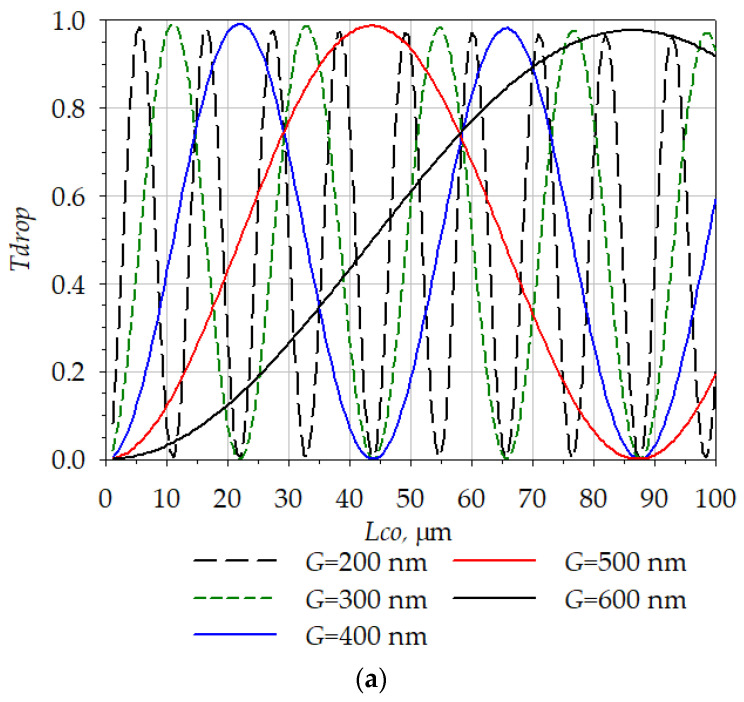
Dependence of the optical transmission coefficient on the coupling length between the waveguides for the OMT with different-plane waveguides (Scheme 8). (**a**) Waveguides with dimensions of 300 × 1000 nm; (**b**) Waveguides with dimensions of 300 × 750 nm.

**Figure 12 micromachines-14-00802-f012:**
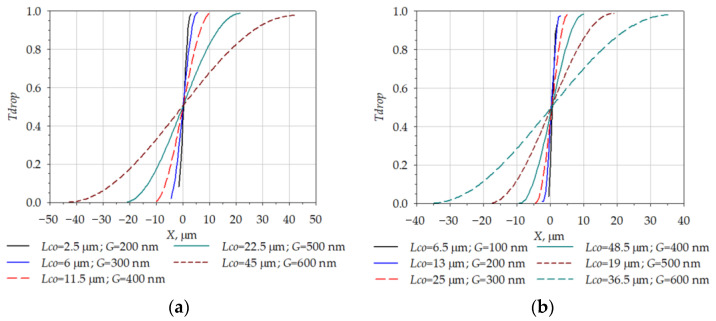
Dependence of the optical transmission coefficient on the inertial mass linear displacement along axis *X* for the OMT with different-plane waveguides (Scheme 8). (**a**) Waveguides with dimensions of 300 × 1000 nm; (**b**) Waveguides with dimensions of 300 × 750 nm.

**Figure 13 micromachines-14-00802-f013:**
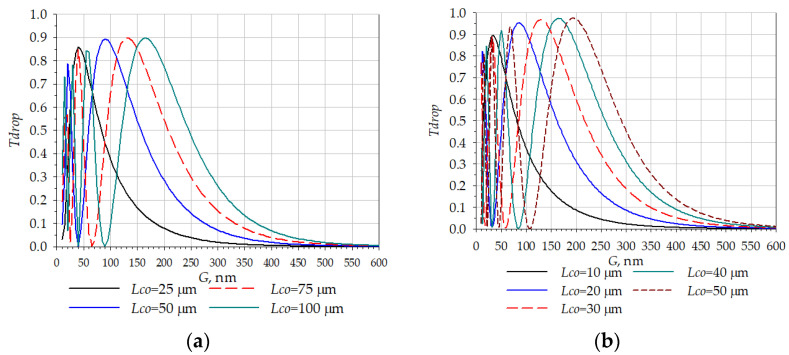
Dependence of the optical transmission coefficient on the gap change for the OMT with single-plane waveguides (Scheme 1, movement along axis Y). (**a**) Waveguides with dimensions of 300 × 1000 nm; (**b**) Waveguides with dimensions of 300 × 750 nm.

**Figure 14 micromachines-14-00802-f014:**
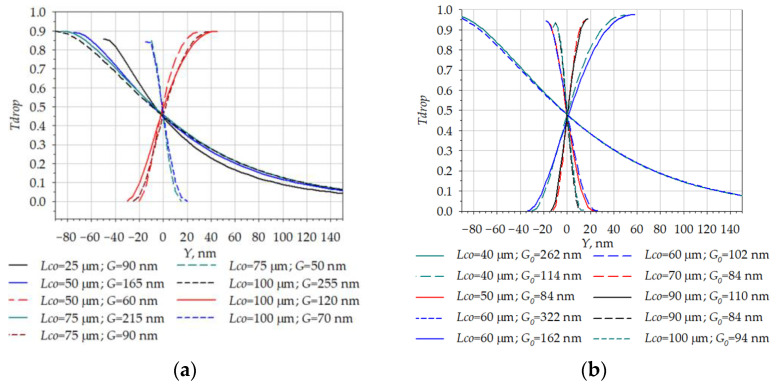
Dependence of the optical transmission coefficient on the inertial mass linear displacement along axis *Y* for the OMT with single-plane waveguides (Scheme 1). (**a**) Waveguides with dimensions of 300 × 1000 nm; (**b**) Waveguides with dimensions of 300 × 750 nm.

**Figure 15 micromachines-14-00802-f015:**
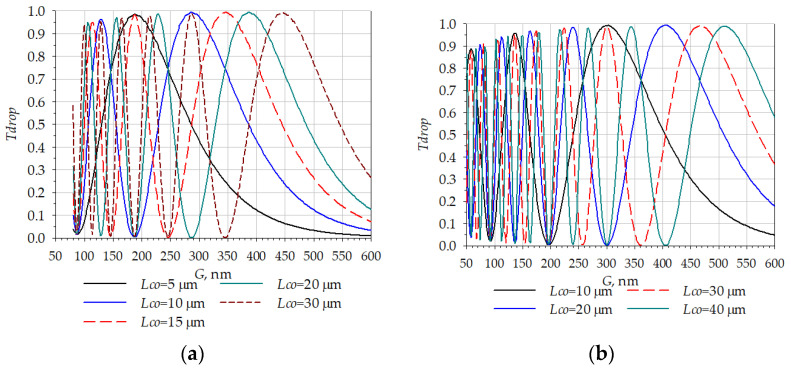
Dependence of the optical transmission coefficient on the gap for the OMT with different-plane waveguides (Scheme 9, movement along axis Z). (**a**) Waveguides with dimensions of 300 × 1000 nm; (**b**) Waveguides with dimensions of 300 × 750 nm.

**Figure 16 micromachines-14-00802-f016:**
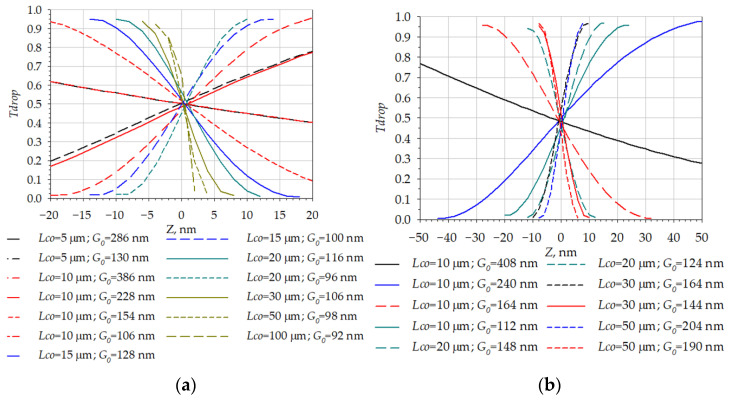
Dependencies of the optical transmission coefficient on the inertial mass linear displacement along axis *Z* for the OMT with different-plane waveguides (Scheme 9). (**a**) Waveguides with dimensions of 300 × 1000 nm; (**b**) Waveguides with dimensions of 300 × 750 nm.

**Figure 17 micromachines-14-00802-f017:**
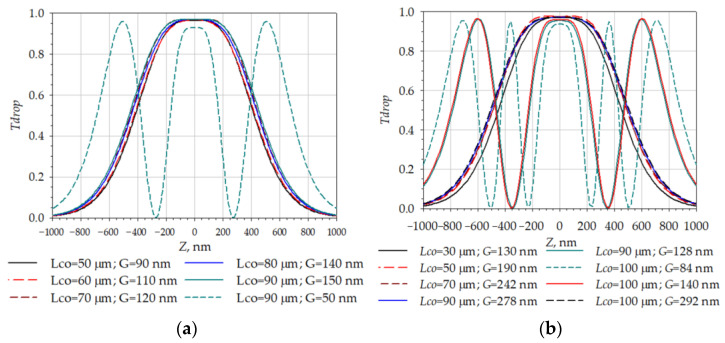
Dependencies of the optical transmission coefficient on the inertial mass linear displacement along axis *Z* for the OMT with single-plane waveguides (Scheme 3). (**a**) Waveguides with dimensions of 300 × 1000 nm; (**b**) Waveguides with dimensions of 300 × 750 nm.

**Figure 18 micromachines-14-00802-f018:**
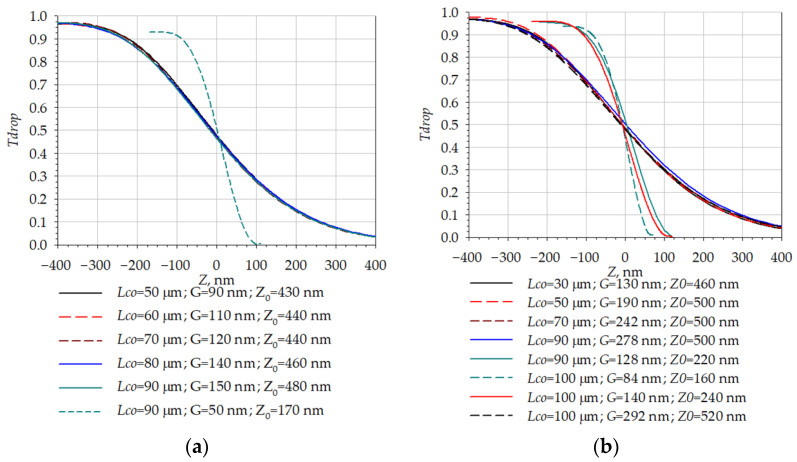
Dependencies of the optical transmission coefficient on the inertial mass linear displacement along axis *Z* for the OMT with single-plane waveguides (Scheme 3). (**a**) Waveguides with dimensions of 300 × 1000 nm; (**b**) Waveguides with dimensions of 300 × 750 nm.

**Figure 19 micromachines-14-00802-f019:**
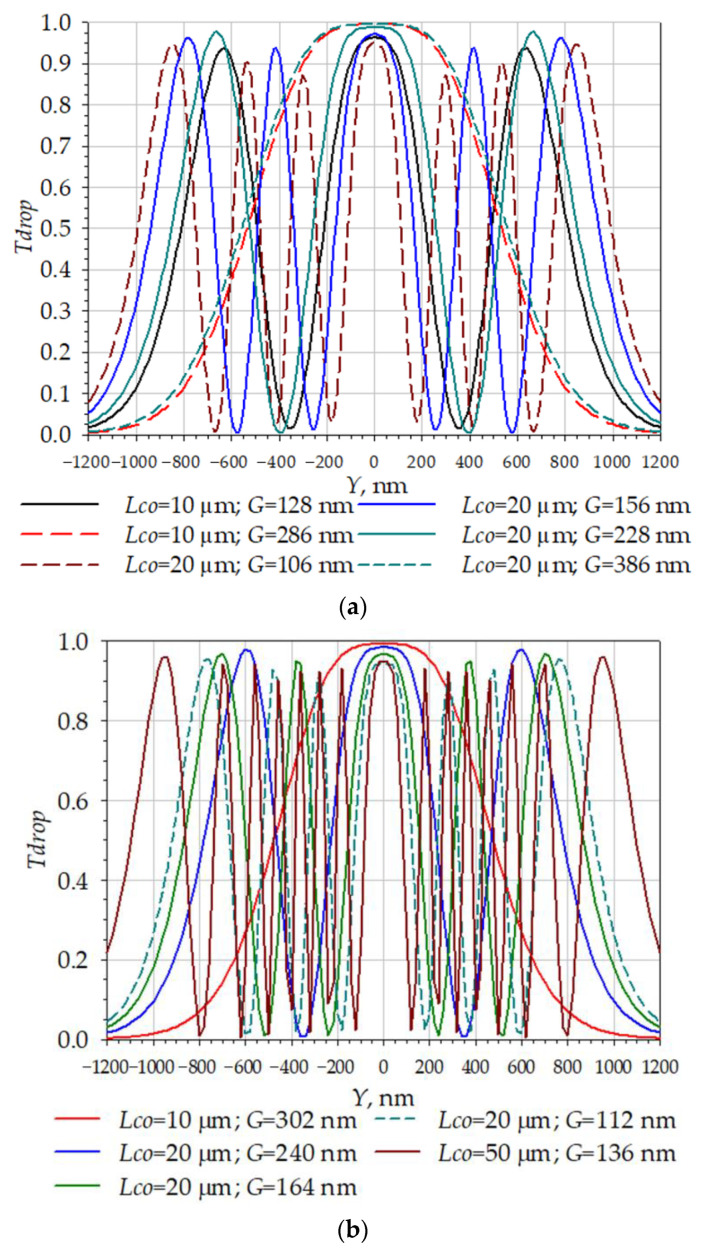
Dependencies of the optical transmission coefficient on the inertial mass linear displacement along axis *Y* for the OMT with different-plane waveguides (Scheme 7). (**a**) Waveguides with dimensions of 300 × 1000 nm; (**b**) Waveguides with dimensions of 300 × 750 nm.

**Figure 20 micromachines-14-00802-f020:**
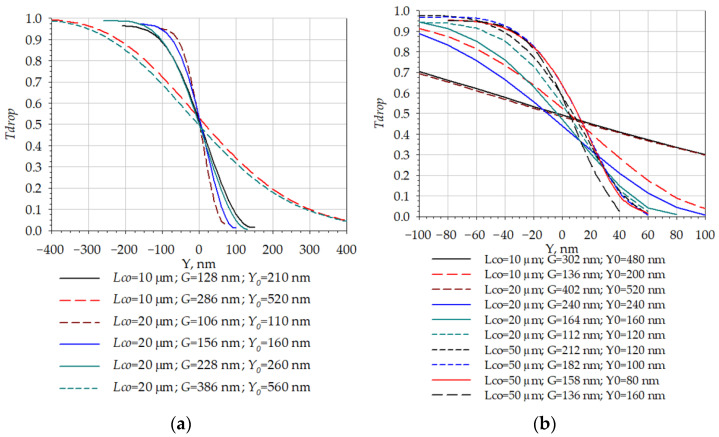
Dependencies of the optical transmission coefficient on the inertial mass linear displacement along axis *Y* for the OMT with different-plane waveguides (Scheme 7). (**a**) Waveguides with dimensions of 300 × 1000 nm; (**b**) Waveguides with dimensions of 300 × 750 nm.

**Table 1 micromachines-14-00802-t001:** Summarized literature review.

Device Type	Sensitivity	Resonance Frequency [Hz]	Self-Noise	Dynamic Range [g]	Bandwidth [Hz]
Michelson interferometer [2]	3.638 nm/g	1742.2 Hz	–	±500	–
Photonic crystal zipper [4]	10 mg/√Hz	–	–	–	20,000
Subwavelength grating pair [3]	1.56 nm/mg	–	–	–	–
Fiber Bragg gratings [5]	0.997 V·g	–	–	–	–
Fiber Bragg grating [16]	14.4–7.5 pm/g	444–940 Hz	15 ng/√Hz	0.2–20	400–900
Optical polymers waveguides [20]	34.1 μm/g	–	–	–	–
On–chip optical interferometry [22]	24.4 µg/√Hz	4500–6400	43.7 ng/√Hz	–	–
Fabry-Pérot resonator [27]	12.5 µW/g	1872	–	±1	–
Fabry–Pérot interferometer [28]	1.022–1.029 mV/(m/s^2^)	1274	–	–	–
Hemispherical optical cavity [29]	1 µg/√Hz	>30,000	–	–	–
Mach–Zehnder interferometer [30]	–	646.56	7.8 × 10^−5^ (m/s^2^)/Hz	–	–
Fibre cantilever [31]	–	–	~0.2 g	–	10–2000
Fiber Bragg grating [32]	450 pm/g	–	–	–	–
Optical microring resonator [36]	31 pm/g	–	–	±7	–
Resonant optical tunneling effect [37]	9 pm/g	–	–	±130	10–1500
Mach–Zehnder interferometer [38]	111.75 mW/g	–	–	–	–
Optical tunneling effect [39]	3 dB/g	–	–	1–10	–

**Table 2 micromachines-14-00802-t002:** Summarized data for the functional schemes of the accelerometer.

Scheme	Altering Parameter	Waveguide Dimensions [nm]	Sensitivity [m^−1^]	Dynamic Range [nm]
	Same–plane waveguides			
1, 4	gap	300 × 1000	(6.25–25) × 10^6^	±20–80
		300 × 750	(6.25–50) × 10^6^	±15–80
2, 5	coupling length	300 × 1000	(12.5–33) × 10^3^	±15,000–40,000
		300 × 750	(10–83) × 10^3^	±6000–40,000
3.6	waveguide overlapping	300 × 1000	(1.25–5) × 10^6^	±100–400
		300 × 750	(1.25–10) × 10^6^	±100–400
	Different–plane waveguides			
9, 12	gap	300 × 1000	(6.25–250) × 10^6^	±2–80
		300 × 750	(6.25–160) × 10^6^	±6–80
8, 11	coupling length	300 × 1000	(1.25–250) × 10^3^	±3000–40,000
		300 × 750	(1.25–500) × 10^3^	±1000–40,000
7, 10	waveguide overlapping	300 × 1000	(1.25–10) × 10^6^	±60–400
		300 × 750	(1.25–12.5) × 10^6^	±40–400

## Data Availability

The data presented in this study are openly available in FigShare at https://doi.org/10.6084/m9.figshare.22340416, accessed on 30 March 2023.

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
