# Peer review of "An Optical Measuring Transducer for a Micro-Opto-Electro-Mechanical Micro-g Accelerometer Based on the Optical Tunneling Effect"

_micromachines, 2023, doi:10.3390/mi14040802_

Round 1
Reviewer 1 Report
In the manuscript of micromachines-2301975, the authors analysed and discussed different schemes for design of MOEM-accelerometer based on a spring mass and an tunneling effect-based optical sensing system based on an optical directional coupler with the following changes to the optical system: gap, coupling length, overlapping area between the movable and fixed waveguides. This research may have guiding significance to the engineering, however, the manuscript should be modified by the following comments.
1. In section of introduction, the novelty of this research should be clarified deeply.
2. Offer a performance comparison table with other typical design schemes or methods of MOEM-accelerometer in detail.
3. Extensive editing of English language and style are required.
4. The number of significant bits of data should be checked carefully.
5. There are two sets of scheme 1~6 in fig.4 and fig.5, which is confusing. Please check it carefully.
6. From the viewpoint of device fabrication, the rationality and difficulty of the design schemes should be discussed deeply.
Author Response
Thank you for your objective peer-review of our article and the comments provided, which allowed us to improve the presentation of our research results.
- In section of introduction, the novelty of this research should be clarified deeply.
- Offer a performance comparison table with other typical design schemes or methods of MOEM-accelerometer in detail.
We added Table 1 and substantiation of novelty in the end of article introduction.
The novelty of the study is to compare different functional schemes of the MOEM-accelerometers featuring optical measuring transducers based on the directional coupler and the optical tunneling effect.
- Extensive editing of English language and style are required.
We have revised the style and grammar of the text. We hope it got better.
- The number of significant bits of data should be checked carefully.
If you mean by significant bit – the number of decimal places, in this study the degree of accuracy we have chosen is sufficient for comparison with each other and this does not affect the conclusions presented.
- There are two sets of scheme 1~6 in fig.4 and fig.5, which is confusing. Please check it carefully.
Agree with the remark. In Figure 5, there was an incorrect numbering of schemes. We fixed scheme numbering.
- From the viewpoint of device fabrication, the rationality and difficulty of the design schemes should be discussed deeply.
We have added an extended explanation to the text in response to your comment. From the perspective of fabrication, the MOEMS-accelerometers with different-plane waveguides are more complex, because they require more technological operations, including complicated lift-off process. This reduces the percentage of usable samples and increases the final cost of the device.
Reviewer 2 Report
Literature review search is good to show some specific data compared with proposed approach in Introduction section. English grammar looks good. Figure quality looks fine. Simulated analysis for waveguides according to various Lco and g looks promising results. Therefore, I can recommend the manuscript could be minor revision if authors can answer the questions as below.
1. How to obtain Equation (1) ? If authors made that, please derive that Equation.
2. What is the unit for Tdrop in Figure 9 ?
3. In Figure 11, there are too many G so authors had better increase the Figure size.
4. In Figure 19, there are too many G and Lco so authors had better increase the Figure size.
5. Please cite the sentence (The principles of optical measurement of displacement~) with ref. (https://www.mdpi.com/2076-3417/11/13/6203).
6. Future work in Conclusion section might be needed.
7. No data availability section. No acknowledgement section.
8. Which kinds of the simulation tool authors used for Figures 9-20 ?
Author Response
- How to obtain Equation (1)? If authors made that, please derive that Equation.
We added the derivation of the equation (1). Equation (1–2) in the article.
- What is the unit for Tdrop in Figure 9?
Optical transmission coefficient (Tdrop) is relation of transmitted optical power (Pdrop) at the output of the drop port to input power (Pinput) and has no units. The equation (9) and an explanation have been added to the text.
- In Figure 11, there are too many G so authors had better increase the Figure size.
We agree. We did not reduce the amount of data, but increased the size of Figure 11.
- In Figure 19, there are too many G and Lco so authors had better increase the Figure size.
We agree. We did not reduce the amount of data, but increased the size of Figure 19.
- Please cite the sentence (The principles of optical measurement of displacement~) with ref. (https://www.mdpi.com/2076-3417/11/13/6203).
We have added a link to the source.
- Future work in Conclusion section might be needed.
We added a few sentences about future work in Conclusion.
- No data availability section. No acknowledgement section.
We change the article structure and added Data Availability Statement.
According to the rules for publishing articles in Micromachines, the Acknowledgments section is optional. All those who participated in the research are co-authors of this work.
- Which kinds of the simulation tool authors used for Figures 9-20?
The calculations made in COMSOL Optics software package using the Finite-Difference Eigenmode (FDE) solver. We added this information in the article.
Round 2
Reviewer 1 Report
The authors have addressed all my comments very well, so I recommend the current version to be published in the journal of Micromachines.